# Corticomotor Plasticity Underlying Priming Effects of Motor Imagery on Force Performance

**DOI:** 10.3390/brainsci12111537

**Published:** 2022-11-13

**Authors:** Typhanie Dos Anjos, Aymeric Guillot, Yann Kerautret, Sébastien Daligault, Franck Di Rienzo

**Affiliations:** 1Laboratoire Interuniversitaire de Biologie de la Motricité, Univ Lyon, Université de Lyon, Université Claude Bernard Lyon 1, EA 7424, CEDEX, F-69622 Villeurbanne, France; 2Allyane^®^, 84 quai Joseph Gillet, 69004 Lyon, France; 3Institut Universitaire de France, F-75000 Paris, France; 4CAPSIX, 69100 Villeurbanne, France; 5Centre de Recherche Multimodal et Pluridisciplinaire en Imagerie du Vivant (CERMEP), Department of Magnetoencephalography, F-69500 Bron, France

**Keywords:** mental imagery, strength, rehabilitation, conditioning, neuromuscular

## Abstract

The neurophysiological processes underlying the priming effects of motor imagery (MI) on force performance remain poorly understood. Here, we tested whether the priming effects of embedded MI practice involved short-term changes in corticomotor connectivity. In a within-subjects counterbalanced experimental design, participants (*n* = 20) underwent a series of experimental sessions consisting of successive maximal isometric contractions of elbow flexor muscles. During inter-trial rest periods, we administered MI, action observation (AO), and a control passive recovery condition. We collected electromyograms (EMG) from both agonists and antagonists of the force task, in addition to electroencephalographic (EEG) brain potentials during force trials. Force output was higher during MI compared to AO and control conditions (both *p* < 0.01), although fatigability was similar across experimental conditions. We also found a weaker relationship between *triceps brachii* activation and force output during MI and AO compared to the control condition. Imaginary coherence topographies of alpha (8–12 Hz) oscillations revealed increased connectivity between EEG sensors from central scalp regions and EMG signals from agonists during MI, compared to AO and control. Present results suggest that the priming effects of MI on force performance are mediated by a more efficient cortical drive to motor units yielding reduced agonist/antagonist coactivation.

## 1. Introduction

The relationship between the cognitive processes mediating the internal representation of actions, i.e., motor cognition, and the physical execution of corresponding movements, has been the focus of a large number of scientific investigations [1,2,3,4,5]. Motor imagery (MI) is the process through which an individual mentally represents an action, without physically executing it. MI requires voluntary recall of visual and/or sensory information from procedural memories, and thus represents a “top-down” process [6]. Action observation (AO) also involves the mental representation of actions, but represents a “bottom-up” form of motor cognition derived from visual inputs to the central nervous system [6]. Abundant data attest the recruitment of brain motor system structures during both AO and MI [3]. While MI is considered an active/voluntary form of motor cognition, it remains subjected to inter-individual variability with regards to MI ability. By contrast, AO is considered a more controllable from of motor cognition since it is driven by exposure to external stimuli [6,7,8]. Motor representations during AO are mediated by the mirror neurons system, which can capture relevant cinematic features of observed actions and mediate both the recognition of others’ actions and motor learning through imitation [9,10].

There is compelling behavioral evidence that combining AO and/or MI with physical practice (PP) promotes motor learning and recovery [11,12,13]. At the fundamental level, functional brain imaging studies have demonstrated that training with AO and MI practice can leverage experience-based plasticity in cortical structures. Brain reorganizations after AO or MI training interventions were comparable to those elicited after training with physical practice [3,14,15,16], which is congruent with the postulate of the neurofunctional equivalence hypothesis between overt and covert action stages [8,17,18]. Pascual-Leone et al. [19] provided the first scientific evidence for parallel cortical reorganization after PP and MI training. Experiments later accumulated evidence that MI training is likely to elicit structural and/or functional reorganizations leading to large-scale functional remapping of the motor networks [20,21]. Likewise, a large number of studies showed the efficacy of motor learning through AO [22,23]. Training with AO contributes to enhance the recruitment of the mirror neurons system, yielding experience-based plasticity. At the structural level, AO practice can lead to structural reorganization of gray matter volumes, facilitating motor skill improvement [24,25].

AO and MI training also contribute to elicit force gains. AO was shown to improve the isometric force of finger muscles by 30% [26], acting as priming stimulus facilitating retrieval of motor coordination [27]. Various MI training interventions, scheduled over several weeks, likewise elicited maximal isometric force gains ranging from 10 to 30% for distal and proximal muscles of the upper limb [28,29]. There is a consensus that force gains as a result of MI training interventions are of central origin, since these were observed in the complete absence of muscle hypertrophy [30,31]. Di Rienzo et al. [32] addressed the short-term efficacy of MI on maximal isometric force. They found that embedded MI practice during the inter-trial recovery periods yielded immediate marginal improvements in force performances of elbow flexor muscles. These short-term effects were interpreted as the result of increased cortical gain over motor units, a postulate well-substantiated by the priming theory of motor improvement through mental forms of practice [33]. The hypothesis that the priming effects of MI on force performance accounts for short-term corticomotor plasticity is receiving increased attention [34]. For instance, recent investigations with transcranial magnetic stimulation confirmed that MI elicits short-term improvements in force performance through increased cortico-spinal facilitation associated with decreased pre-synaptic inhibition of alpha motor neurons [35]. Yet, a major problem is the scarcity of studies addressing the neurophysiological underpinnings of short-term force gains in response to embedded motor cognition interventions.

The aim of this study was to expand current understandings of the neurophysiological processes underlying priming effect of MI on maximal isometric force performances. The state of the art suggests that force gains result from increased cortical gain over motor units [30,31,32,36]. This hypothesis is derived from the association between increased cortical output and increased force as a result of MI training interventions [30,31]. It was, however, never addressed experimentally from connectivity measures between central nervous system activity and peripheral motor responses. Accordingly, we assessed corticomotor connectivity patterns in response to embedded MI training interventions known to elicit priming effects on maximal isometric force performance. We also tested, for the first time, the priming effect of embedded AO on maximal isometric force. We hypothesized greater priming effects following MI compared to AO, due to its top-down nature. We finally expected distinct patterns of corticomotor connectivity between the two forms of motor cognition. Addressing the aforementioned experimental hypotheses should contribute to deepen current understandings of the priming effects of motor cognition on force performance.

## 2. Materials and Methods

### 2.1. Participants

Twenty participants (12 males, 8 females) aged from 19 to 39 years (25.9 years ± 4.68) volunteered to participate in the experiment. All met the following inclusion criteria: (i) age 18–45 years old, (ii) right-handedness according to the short form of the Edinburgh Handedness inventory [37], (iii) regular practice of physical and/or sporting activities at a recreational level (>2 sessions of 1 h per week over the last 6 months), (iii) body mass index ranging 19–24. Exclusion criteria were (i) history of neurological/psychiatric or locomotor disorders, (ii) diagnosis of chronic disease(s) related to the metabolic syndrome (e.g., hypertension, diabetes and obesity), (iii) being involved in a resistance training program targeting the upper limbs at the time of the experiment. The study was ethically approved by the review board of the University. Each participant provided a written informed consent form before enrolling in the study, in keeping with the principles and statements laid out in the Declaration of Helsinki (2013). Prior to the experiment, participants completed the Functions of Observational Learning Questionnaire (FOLQ; Cumming et al., 2005) and the Movement Imagery Questionnaire-3 (MIQ-3; Williams et al., 2012). These are qualitative tools investigating the ease of AO and MI vividness, respectively. The FOLQ specifically investigates two cognitive and one motivational functions of AO using a 7-point Likert-type scale (1: « I never use »; 7: « I often use »). Cumming et al. (2005) reported high internal consistency with Cronbach’s alphas ranging from 0.84 to 0.90. The MIQ-3 is a 12-item questionnaire requiring participants to imagine performing different movements from internal, external, and kinesthetic perspectives on a 10-point Likert-type scale (1: “Very difficult to see/feel”; 5: “Very easy to see/feel”). Internal consistency of MIQ-3 is generally deemed acceptable with Cronbach’s alphas >0.80 [38].

### 2.2. Experimental Procedures

#### 2.2.1. Experimental Design

The design consisted of three successive experimental sessions scheduled within a span of 3 weeks (Figure 1a). Experimental sessions were separated by 48–96 h to prevent carryover effects (e.g., residual fatigue from one session to another). Experimental sessions lasted about 45 min, including preparation time, and consisted of a force training session where participants completed maximal elbow flexions against an immovable force platform [34] (Figure 1b). Participants completed 10 trials with their dominant (right) arm, immediately followed by 3 trials with their non-dominant (left) arm. Indeed, we were interested in potential sparing effects on force performance of the contralateral limb. Increased motor output from the untrained limb following a period of unilateral training through PP is referred to as “cross education” [39]. Cross-education typically reflects central nervous system adaptations [39,40]. Activation of contralateral Ia afferents and alpha motor neurons from the contralateral limb might modulate ipsilateral reciprocal inhibitory pathway, hence affecting the ability to generate force [41]. For each trial, participants were requested to sustain their effort for 12 s. Force trials were separated from each other by 1 min rest periods allocated to recovery (Figure 1a). During recovery periods, we administered three distinct experimental conditions, i.e., one per session. During the first condition, participants were requested to complete MI of the maximal isometric contraction of elbow flexor muscles (MI condition). They performed MI with their eyes closed, and were instructed to combine visual and kinesthetic modalities according to the following script: “Imagine yourself lifting the force platform with all your force. Feel the intense contraction of your biceps and your shoulder muscles during the sustained maximal contraction. Focus on the total recruitment of muscle fibers throughout the duration of the effort”. They completed 3 MI trials of 12 s during each 1-min inter-trial rest period. During the second condition, they passively watched a video of a bodybuilder athlete performing the maximal isometric force task for 12 s, according to a sagittal view, with his upper limb muscles visible throughout the trial (AO condition). They completed 3 AO trials during the 1-min inter-trial rest periods. During the control condition, participants passively watched 3 times a 12 s video of basketball shooting by professional athletes. To prevent order bias, experimental conditions were administered in a counterbalanced order across participants throughout experimental sessions (block randomization).

#### 2.2.2. Experimental Settings

Experimental settings replicated and followed the methodology used by Di Rienzo et al. [32,34]. Participants sat on a bench equipped with an adjustable reclining seatback. We used a goniometer to ensure a 90° elbow flexion. Hand position was rigorously controlled by measuring that the wrist, hand palm and fingers were in flat contact with the reversed force platform. During isometric elbow flexions, participants stared at a cross-mark fixed at eye height. For a force trial to be declared valid, participants had to maintain for 12 s a standardized position requiring a permanent contact between the reclining seatback and (i) the back of their head, (ii) the posterior apex of the thoracic spine, and (iii) the pelvis. This was intended to prevent compensatory trunk movements. Throughout the study, the timing of maximal isometric force contractions and inter-trial recovery periods was externally cued using a stimulus delivery/experiment control software designed for purposes of neuroscience research (Presentation^®^, Version 18.0, Neurobehavioral Systems, Inc., Berkeley, CA, USA, http://www.neurobs.com (accessed on 7 October 2022)).

### 2.3. Dependent Variables

#### 2.3.1. Force Performance

The elbow flexion force was measured with a force platform (AMTI, model OR6-7-2000, Watertown, MA, USA). Data were continuously recorded and synchronized by LabChart Pro V8© (ADInstruments Pty Ltd., Dunedin, New Zealand, 2014) at 1000 Hz. After performing frequency and residual analyses on raw signals, data were smoothed with a zero-lag 4th-order low-pass Butterworth filter, with a 20 Hz cut-off frequency. During each trial, the sudden force increase in response to the auditory stimulus was detected using a threshold detection function (Matlab^®^, R2021b, Natick, Massachusetts, The MathWorks Inc.). The total force was calculated by integrating the force slope with respect to the duration of each trial (12 s, trapezoid rules).

#### 2.3.2. Electromyography

Surface electromyograms (EMG) were collected from the *biceps brachii*, *anterior deltoideus*, and *triceps brachii* using pairs of surface electrodes (1 cm EMG Triode, nickel-plated brass, inter-electrode distance 2 cm, Thought Technology, Montreal, Canada). These muscles had, respectively, agonist, agonist/synergist and antagonist functions during the force task. After shaving and cleaning the skin with alcohol, EMG sensors were positioned according to usual recommendations of the surface electromyography for the non-invasive assessment of muscles (i.e., SENIAM, [42]. Electrode location was marked with a pen and photographed to ensure reproducible positioning across experimental sessions. Raw signals were collected using the TrignoTM Wireless EMG© system (2014, Delsys Incorporated, Natick, Massachusetts, United States of America), which allows continuous synchronization of EMG data using LabChart ProV8© (ADInstruments Pty Ltd., Dunedin, New Zealand, 2014). To index muscle activation, we applied a root mean square filter on the raw EMG signals across the time window corresponding to maximal isometric force trials of the warm-up and experimental sessions (EMG_RMS_).

#### 2.3.3. Electroencephalography

1.Data acquisition

Spontaneous electrical brain activity (EEG) was recorded using 16 Ag-AgCl electrodes mounted in an elastic cap at C3, C4, Cp3, Cp4, Cp8, Cz, F3, F4, Fp1, Fp2, Fz, P3, P4, and Tp7 locations of the international 10–20 system (MLAEC2 Electro-cap system 2, ADInstrument^®^, Dunedin, New Zealand). Two mass electrodes were positioned on the right internal canthus and the right acromion. The EEG signal was filtered at 0.5–200 Hz and sampled at 250 Hz. Electrode impedance was kept homogeneously lower than 5 kΩ. An additional 50 Hz hardware notch filter was applied to avoid power line contamination. The electrodes were connected to the input box of the EEG recording system, i.e., two Octal Bio Amps (FE238, Octal Bio Amp, ADInstrument^®^, Dunedin, New Zealand) and one Power Lab 16/35 data acquisition system (PL3516, Power Lab 16/35, ADInstrument^®^, Dunedin, New Zealand).

2.Pre-processing

EEG data was processed using Brainstorm [43], which is documented and freely available under the GNU general public license (https://neuroimage.usc.edu/brainstorm (accessed on 1 January 2020)). EEG signals were first referenced relative to the average reference electrode. Eye blink artifacts were corrected with a signal space projection algorithm [44]. Considering that EEG signals were collected during maximal voluntary contractions, we corrected for muscle artifacts in EEG signals using an empirical mode decomposition [45]. This removed residual muscle artifacts from the multichannel EEG signals. We further applied a band pass filter to the signal [0.5–200] Hz. We were interested in between-condition differences in corticomotor connectivity patterns during maximal voluntary isometric contractions. EEG signals were thus epoched in [−5, 10] s time windows relative to the onset of each maximal isometric force trial. Epochs were visually controlled for artifacts (15.4% of trials were rejected over 17 subjects), and EEG channels containing artefacts were excluded from further analyses (less than 15% channel rejected per subject).

3.Coherence analysis

Corticomotor connectivity (CMC) analyses provide reliable insights on the cortical regulation of muscle activation, which differs according to the functional role of each muscle [46]. To examine functional coupling between EEG and EMG signals, we first rectified the raw EMG measured from the *biceps brachii*, *anterior deltoideus*, and *triceps brachii*. We then computed the 1 (EMG_RECTIFIED_) * N (EEG sensors) imaginary coherence matrices over the (0, 10) s time window corresponding to each maximal isometric force trial (https://neuroimage.usc.edu/brainstorm/Tutorials/CorticomuscularCoherence, accessed on 1 January 2020). Coherence matrices were averaged across trials and experimental conditions, yielding for each muscle a coherence power spectrum across the 0.5–60 Hz frequency range. This enabled us to disclose the frequency windows exhibiting imaginary coherence peaks, which were considered the frequency domains of interest for the study of CMC across experimental conditions. This data-driven approach controlled for selection bias. Within the frequency window(s) of interest, we examined the standardized (Z-score) imaginary coherence topographies in the EEG sensors-space for all muscles and experimental conditions. Z-scores provide estimates of effect size under the assumption of normal distribution [47]. We also calculated a CMC ratio corresponding to the normalized difference in EEG/EMG imaginary coherence between agonist (i.e., *biceps, brachii,* and *anterior deltoideus*) and antagonist (i.e., *triceps brachii*) muscles, following Equation (1).
(1)$${\mathrm{CMC}}_{\mathrm{RATIO}}=\frac{\left({\mathrm{CMC}}_{\left(biceps\;brachii\;and\;anterior\;deltoideus\right)}-{\mathrm{CMC}}_{\left(triceps\;brachii\right)}\right)}{\left({\mathrm{CMC}}_{\left(biceps\;brachii\;and\;anterior\;deltoideus\right)}+{\mathrm{CMC}}_{\left(triceps\;brachii\right)}\right)}$$

The CMC_RATIO_ was used as a standardized connectivity index to study agonist/antagonist CMC patterns across experimental conditions.

#### 2.3.4. Self-Report Ratings

After each experimental session, participants rated on a 10-point Likert-type scale (1: “Very low”; 10: “Very high”) their motivation to engage in the experimental session, and their perceived performance across maximal isometric force trials. They also rated the perceived difficulty to complete the recovery condition for each experimental session. They also reported their perceived MI vividness on a 10-point Likert-type scale (1: “Absence of sensations, only thinking about the movement”; 10: “Identical sensations as during physical practice of the task”) after the MI experimental condition.

### 2.4. Statistical Analyzes

#### 2.4.1. Linear Mixed-Effects Analysis

We used R [48] and nlme [49] to run a linear mixed-effect analysis of the dependent variables. We built a series of regression models with by-subjects random intercepts. We first analyzed total force data collected during the maximal isometric force trails of the warm-up, using the fixed of condition (MI, AO, control). Then, total force measures collected during the experimental sessions were analyzed using the fixed effect of condition and its interaction with the fixed effects of TRIAL (numeric regressor, 1 to 10) and EMG_RMS_ from the *biceps brachii*, *anterior deltoideus*, and *triceps brachii*. CMC_RATIO_ was analyzed by testing the fixed effects of condition, EEG channels and muscle (*biceps brachii*, *anterior deltoideus*, *triceps brachii*), for each frequency domain(s) of interest revealed by the power spectrum analysis. Standardized questionnaires scores were investigated using simple linear regression with the categorical variable dimension (FOLQ: strategy, skill, performance; MIQ-3: external, internal, kinesthetic) as predictor. Eventually, self-reports of perceived performance were analyzed using the fixed effect of condition (MI, AO, control). Visual inspection of residual plots did not reveal any obvious deviations from homoscedasticity or normality. The alpha threshold for the type 1 error rate was set at 5%. As effect sizes, we reported estimates of how much variance of the predicted variable was accounted for by main/interaction effects of the model (partial Cohen’s F), using the *ad-hoc* procedure for linear mixed effects models implemented in the effectsize package [50,51]. Main and interactions effects were investigated post-hoc using general linear hypotheses testing of planned contrasts from the multcomp package [52,53]. We applied Holms’ sequential Bonferroni corrections to control the false discovery rate.

#### 2.4.2. Power Considerations

A priori calculations to determine the sample size were conducted using R [48] and the package pwr [54]. Sample size should afford adequate statistical power (p_1−β_) to ensure reproducibility of the findings. Within-subject designs typically require a reduced number of participants compared to between-subject designs in order to detect systematic variations affecting the dependent variables [55,56]. Consistent with earlier work by our team [34], we found that a sample of 20 participants yielded a statistical power of p_1−β_ > 0.80 to detect effect sizes >2% of explained variation for the main effect of condition on total force. Additionally, we reported a posteriori power calculations for statistically significant main/interactions effects.

## 3. Results

### 3.1. Force Performance Analysis

#### 3.1.1. Baseline Levels

Condition did not affect maximal isometric force data collected during the warm-up trials of the design, for both trained and untrained arms (all *p* > 0.05, Table 1). Likewise, there were no differences in EMG_RMS_ recorded from the *biceps brachii*, *triceps brachii*, and *anterior deltoideus* for trained and untrained arms (all *p* > 0.05, Table 1). Accordingly, we used raw total force and EMG_RMS_ data collected during the experimental sessions for the forthcoming steps of data analysis.

#### 3.1.2. Maximal Isometric Force during Experimental Sessions

For the dominant trained arm, the linear mixed effect analysis carried on total force data revealed a two-way interaction between condition and EMG_RMS_ recorded from the *triceps brachii* (F(2, 435) = 9.84, *p* < 0.001, Cohen’s F = 0.21, p_1−β_ = 0.97). As shown by Figure 2b, the positive predictive relationship between EMG_RMS_ recorded from the *triceps brachii* and the total force was greater during control compared to both MI (+82.07 N.mV^−1^ ± 21.00, *p* < 0.001) and AO (+80.04 N.mV^−1^ ± 20.30, *p* < 0.001). We observed a marginal trend for the condition by EMG_RMS_ interaction effect (F(2, 435) = 1.67, *p* = 0.09, Cohen’s F = 0.09, p_1−β_ = 0.45). This originated from a positive relationship pattern between the total force and *anterior deltoïdeus* EMG_RMS_ under AO (+10.26 N.mV^−1^ ± 7.62), whereas a negative relationship pattern was present between the total force and *anterior deltoïdeus* EMG_RMS_ during control (−7.21 N.mV^−1^ ± 5.37). Importantly, we found a main effect of condition (F(2, 435) = 10.61, *p* < 0.001, Cohen’s F = 0.22, p_1−β_ = 0.99). Irrespective of trials, the total force was higher during MI compared to both AO (0.71 N ± 0.19, *p* < 0.001) and control (0.64 N ± 0.20, *p* < 0.01), while there was no difference between AO and control (*p* > 0.05, Figure 2a). EMG_RMS_ from the *biceps brachii* also affected total force values (F(1, 435) = 78.42, *p* < 0.001, Cohen’s F = 0.45, p_1−β_ > 0.99) positively (+4.23 N.mV^−1^ ± 0.61, *p* < 0.001).

For the untrained arm, total force was affected by the main effects of trial (F(1, 133) = 32.58, *p* < 0.001, Cohen’s F = 0.49, p_1−β_ > 0.99) and condition (F(2, 133) = 6.87, *p* = 0.001, Cohen’s F = 0.32, p_1−β_ = 0.93). Trial negatively predicted total force (−0.92 N.mV^−1^ ± 0.16, *p* < 001). Post-hoc analyzes showed that the total force across trials during MI outperformed those during control (1.26 N ± 0.34, *p* < 0.001). The difference between AO and control fell short from the statistical significance threshold (0.74 N ± 0.33, *p* = 0.06, Figure 2b). There was no difference between MI and AO (*p* > 0.05).

### 3.2. Corticomotor Coherence Analysis

Figure 3a shows the CMC power spectrums between 0 and 60 Hz for the *biceps brachii*, *triceps brachii*, and *anterior deltoieus*, averaged across participants and experimental conditions. CMC power spectrums revealed peaks in the alpha (8–12 Hz) frequency band for all muscles (Figure 3a). The alpha rhythm was thus selected as a frequency band of interest. The topographical distribution of imaginary coherence values in the EEG sensors-space across the alpha frequency band revealed that the highest coherence values were located in CZ for the *biceps brachii* during MI, whereas they gravitated around C3 during CONTROL (Figure 3b). Interestingly, the highest coherence values recorded for the *triceps brachii* involved C3 during control, but not during MI and AO. During AO, coherence values between central electrodes of the contralateral hemisphere (i.e., C3, CP3) and *anterior deltoïdeus* EMG were higher than those obtained for the *biceps brachii* EMG, which was not the case during control and MI (Figure 3a).

CMC_RATIO_ topographies revealed higher coherence between central EEG sensors and *biceps brachii/anterior deltoideus* EMG (relative to the coherence between central EEG and *triceps brachii* EMG) during MI compared to AO and control (Figure 3c). CMC_RATIO_ was affected by the two-way interaction between condition and channel (F(12, 559) = 1.80, *p* < 0.05, Cohen’s F = 0.20, p_1−β_ = 0.91). Post-hoc investigations revealed that differences between Cz and C3, and P3 and TP7 electrodes during MI were higher than the corresponding differences found during control (all *p* < 0.05, Figure 3c). Additionally, differences between TP7 and CZ, C3, F3, and FP1 were reduced during AO, compared to that observed during control (all *p* < 0.05).

### 3.3. Analysis of Participant’s Subjective Ratings

#### 3.3.1. Standardized Questionnaires

The linear mixed effects analysis carried on MIQ-3 scores revealed no effect of dimension (F(2, 32) = 1.03, *p* = 0.37). MIQ-3 global score at the group level was 70.88 ± 8.14, indicative of an intermediate level between “Quite easy to visualize” and “Easy to visualize”. The main effect of dimension, however, affected FOLQ scores (F(2, 32) = 16.52, *p* < 0.001, Cohen’s F = 1.02, p_1−β_ = 0.95). FOLQ scores on the performance dimension (3.42 ± 1.40) were lower than for both strategy (4.41 ± 1.35) and skills (5.15 ± 1.05) dimensions (both *p* < 0.01). FOLQ scores for skill were also higher than those for strategy (*p* < 0.05).

#### 3.3.2. Standardized Questionnaires

Self-reports of motivation to complete the experimental sessions and perceived efficacy across force trials were unaffected by the condition (F(2, 37) = 0.61, *p* = 0.54, Cohen’s F = 0.18, p_1−β_ = 0.14; F(2, 37) = 0.90, *p* = 0.42, Cohen’s F = 0.22, p_1−β_ = 0.21, respectively). The perceived difficulty, however, was affected by the main effect of condition (F(2, 37) = 6.13, *p* = 0.005, Cohen’s F = 0.58, p_1−β_ = 0.78). Participants perceived a greater difficulty to complete the MI inter-trial recovery condition (6.70 ± 2.54) compared to control (4.94 ± 2.46) (*p* < 0.01, Figure 4). However, there was no difference between the perceived difficulty of MI and AO (5.75 ± 2.69) recovery conditions (*p* > 0.05). Mean MI vividness ratings on the 10-point Likert scale were 7.05 ± 2.08.

## 4. Discussion

The present study investigated the short-term effects of AO and MI administered during inter-trial recovery periods of a force training session. We aimed to further elucidate the neurophysiological processes underlying priming effects of MI on maximal isometric force performance [32,34,35]. Past experiments investigated changes in corticospinal excitability in response to MI practice interventions on force performances [35,57]. Here, we focused on the corticomotor connectivity associated with embedded MI and OA during a force training session using EEG-EMG coherence. We first confirmed that embedded MI practice improved maximal isometric force performances of the dominant upper-limb [32,34]. MI also outperformed AO. This represents an original finding, since the priming effects of these two forms of motor cognition on maximal isometric force had not yet been compared within a sole experimental design. Interestingly, the behavioral results were predicted by distinct profiles of muscle activation across experimental conditions, particularly reduced involvement of the *triceps brachii* during both MI and AO compared to control. EEG-EMG imaginary coherence during force trials peaked within the alpha-range (8–12 Hz). Compared to AO and the control condition, CMC patterns during MI involved greater connectivity between the agonists of the force task and EEG signals recorded from central regions of the scalp.

The behavioral analysis of motor performances revealed no difference between AO and control conditions. However, MI outperformed both in terms of maximal isometric force output across trials. Force gains reported in the literature ranged from 2–5% at the single-session level [32], up to 36% after 7 weeks of intervention [58]. In the present study, we found that force increased by 4.3% after a single session, which is in line with previous findings using the same experimental paradigm and confirms the positive short-term influence of embedded MI [34]. Although increases in force performances were previously reported after MI and AO administered in isolation [26,59,60,61], the direct comparison of embedded AO and MI practice revealed here the superiority of MI. The lack of force gains during AO could originate from reduced levels of central nervous system facilitation compared to MI. This postulate is in keeping with the passive nature of AO compared to the voluntary process of action representation. Furthermore, functional brain imaging showed that the patterns of brain recruitment between MI and AO were hierarchically organized, with cerebral recruitment patterns during MI being closer from those elicited during PP [62]. While it was recently established that combined AO and MI yielded increased corticospinal facilitation compared to AO or MI performed alone [63], recent data suggests that MI alone drives most of the facilitation [64]. Henceforth, it is suggested that the superior priming effects of MI on force, compared to AO and the control condition, accounts for a greater degree of central nervous system recruitment under voluntary (top-down) compared to externally-guided (bottom-up) motor simulation. An important caveat to definitive conclusions regarding the superiority of MI compared to AO is the isometric nature of the contractions involved in the force task. Although muscle contractions of the bodybuilder were visible under the AO condition, maximal isometric force exercises may be less prone to the formation of motor memories through AO compared to force exercises that require dynamic contractions. This must be considered a limitation, since testing the effects of AO and MI on performance for force exercises that involve poly-articular coordination through dynamic contractions may have yielded a different results pattern. This is particularly true in novices who may experience difficulties to engage in MI and may benefit from AO during early learning stages [7,65]. Furthermore, we cannot rule out that the use of a male model could have hampered the efficacy of the AO condition in female participants. We hypothesized that the effect of embedded MI practice on maximal isometric force of the dominant arm would transfer to the untrained non-dominant arm. The results confirmed this hypothesis, since MI only outperformed the control condition for non-dominant arm force trials. While there is sporadic evidence for contralateral transfer effects of MI training in force paradigms, these were never reported at the single-session level [36,58,66]. Past experiments demonstrated that a single session of repetitive upper-limb movements was sufficient to leverage neural plasticity in corticospinal tract projections to the contralateral, untrained muscle [67,68]. The transfer of priming the effects of embedded MI practice on maximal isometric force to the untrained arm suggest that comparable results may be achieved through repeated MI practice.

The psycho-neuromuscular theory holds that MI elicits a preliminary facilitation of corticospinal pathways mediating forthcoming performances [69]. This provides a scientific rationale for the priming effects of MI on motor performance [70]. Originally, priming was assumed to be mediated by increased cortical output yielding increased cortical gain over motor units. Recent transcranial magnetic stimulation data indicates that MI also elicits the preliminary activation of low-threshold spinal interneurons involved in the integration of proprioceptive information from Ia afferent fibers, leading to the downregulation of pre-synaptic inhibition of alpha motor neurons [35,71,72]. Priming effects of MI may thus be mediated by both cortical and spinal plasticity. Here, we restricted the scope of our investigations to the cortical component. A limitation is the absence of source reconstruction, which would require a higher number of EEG sensors to control for volume conduction. We hypothesized distinct patterns of connectivity between brain oscillations and peripheral muscle activation during MI compared to the control condition. Present results shed light to a plausible facilitation of intramuscular coordination. Ranganathan et al. [30] provided the first experimental evidence for increased cortical motor potentials amplitudes after a 12-week MI intervention. Increased motor potentials from CZ/C3 were associated with increased maximal isometric force. However, whether such increases underpinned force improvements through differences in agonist/antagonist activation patterns remained unclear. CMC is used to assess neural communication between central and peripheral sensorimotor systems [73,74]. The magnitude of CMC, which indexes the synchronization level between the phases of EEG and EMG signals, is modulated by both the force level and degree of agonist/antagonist coactivation [75,76,77]. Here, the CMC power spectrum revealed higher coherence in the alpha (8–12 Hz) frequency band for all muscles. Although CMC was more frequently described in the beta band (13–30 Hz) during sustained isometric muscle contractions (for a review, see [78]), it is also frequently investigated in the alpha band [79,80]. Cremoux et al. [77] reported greater alpha coherence as a result of short-term motor training. Alpha oscillations thus represent a relevant frequency domain for the study of corticomotor connectivity changes associated with the early stages of training. Alpha CMC topographies revealed increased coherence between the agonists and Cz during MI compared to control. By contrast, coherence peaks between the *biceps brachii* EMG and C3 merged during control. Whether this pattern of results reflects an emphasis on the putative role of pre-central structures in the set-up of inter-muscular coordination during online control of voluntary contractions during MI remains speculative, since the limited number of EEG channels did not allow for source reconstruction. Interestingly, there was a trend for increased involvement of the *anterior deltoïdeus* EMG during force performance during AO compared to MI and control. This trend is consistent with CMC patterns found for the *anterior deltoideus* during AO. Indeed, CMC topographies revealed that CMC_RATIO_ peak in C3 during AO originated from coherence peaks with the *anterior deltoideus*, but not the *biceps brachii*. This pattern was absent during MI and control, where CMC_RATIO_ peaks were observed in CZ and C3, respectively, for the *biceps brachii*. The *anterior deltoideus* is a synergist of the force task. During MI, coherence peaks also emerged between C3 and the *anterior deltoideus* in addition to the peak between Cz and the *biceps brachii*. Considering that force performances were lower during AO than during MI, we suggest that AO-emphasized muscles synergies at the expense of the activation of prime agonists. Specifically, it has been postulated that AO primarily primed intermuscular coordination [34]. This hypothesis seems congruent with the role of AO in the development of sequencing and timing of basic action concepts [81]. 

Our present EMG and CMC data suggest that short-term force improvements elicited by embedded MI practice originate from decreased agonist/antagonist coactivation. This interpretation is congruent with the absence of differences in the predictive relationship between force performances and *biceps brachii/anterior deltoïdeus* activation between the MI and control conditions. These muscles are the main agonist muscles involved in the force task. By contrast, the relationship between *triceps brachii* EMG and force performances was reduced during MI. This was associated with a reduced overlap in CMC topographies between the *biceps brachii/anterior deltoideus* and the *triceps brachii* during MI compared to CONTROL, as revealed by CMC_RATIO_ in the EEG sensors-space. These findings corroborate earlier reports of increased torque production of the angle plantar-flexor muscle after 7 weeks of MI training, where force gains were associated with differences in the distribution of activities of muscles with antagonist functions [58]. Noteworthily, the reduction of agonist/antagonist activation is a well-established early neural adaptation to resistance training, and the reduction of the agonist/antagonist coactivation has been suspected to account for force improvements during the early onset of isometric resistance training [82,83,84,85,86]. As reported by Baker [87], EEG-EMG interactions serve a functional role during the control of voluntary actions and cannot be interpreted as a simple propagation of cortical oscillations towards the periphery. The modulation of these interactions indicates specific agonist/antagonist synergies, and could reflect a more refined cortical control of peripheral effectors, as well as increased levels of force production [46,88]. In line with previous findings by Dal Maso et al. [89], CMC magnitude differed between antagonist and agonist muscles, suggesting a specific regulation of muscle activation according to their main function. In light with previous investigations reporting that the training specificity led to different antagonist contributions regarding force production, our results underline that decrease of CMC alpha magnitude in the antagonist muscle could be the reflection of a higher cortical involvement to better perform the task. Our results overall provide a more in-depth understanding of the rapid neurophysiological adaptations underlying short-term force as a result of embedding MI. We suggest that embedded MI practice primes force performance improvements through the early onset of neural adaptations classically observed after physical training. This argues for integrated forms of MI-practice interventions in actual training contexts [90,91].

## 5. Conclusions

To conclude, force gains found in the present study confirm earlier findings on the effects of embedded MI practice on maximal isometric force performances [32]. Results demonstrated that short-term effects of MI during inter-trial recovery periods on maximal isometric force were mediated by neural adaptations, indexed here from differences in CMC patterns. We ruled out a bias in perceived motivation and/or efficacy, since these did not differ between experimental conditions based on subjective measures. The present data points, for the first time, to the regulation of agonist/antagonist co-activation as an underlying mechanism to the short-term effects of embedded MI practice on maximal isometric force. This has important practical implications. It first supports the relevance of embedded MI practice during the early stages of resistance training programs in novice participants. In patients who suffered peripheral injuries, embedded MI practice within rehabilitation sessions engaging affected body parts in voluntary isometric contractions—which present the advantage of reducing the risk of excessive joint solicitation—is also particularly justified. However, further research is required to assess both short-term and long-term neurophysiological implications of such interventions in clinical settings. The present effects of embedded MI practice are also certainly possibly exploitable in rehabilitative and assistive technology design and development. Our findings finally provide original insights on short term cross-education effects in response to embedded MI practice. Cortical adaptations underlying CMC differences at the single MI session level could contribute to activating both hemispheres [92], yielding improved efferent neural drive to the untrained muscles. We did not address this hemisphere activation hypothesis, since we focused experimental hypotheses on the dominant trained limb. While this must be considered a limitation, this also provides a promising perspective for future investigations of short-term neurophysiological adaptations induced by mental training.

## Figures and Tables

**Figure 1 brainsci-12-01537-f001:**
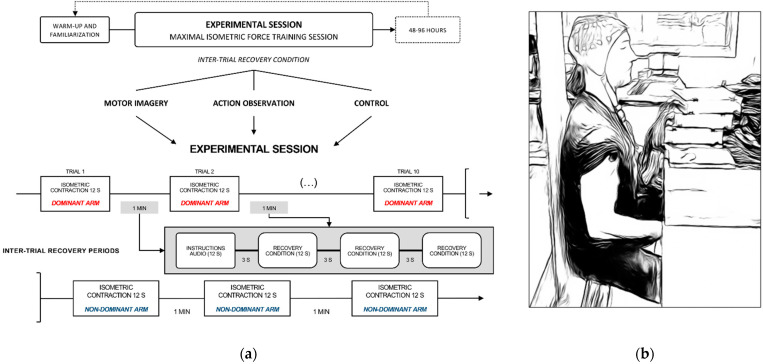
Schematic representation of the experimental design. (**a**) Flowchart depicting the factorial plan of the within-subjects experimental design, which included three experimental sessions. Participants first completed warm-up force trials, followed by a familiarization to the experimental condition of the day. Next, they underwent a strength training session consisting in 10 maximal isometric force trials sustained for 12 s, with 1 min inter-trial rest periods allocated to recovery. Each experimental session involved one of the three possible inter-trials recovery condition (i.e., control, motor imagery and action observation), administered in a counterbalanced order across participants. Each experimental session involved a total of 13 maximal isometric contractions (i.e., 10 with the dominant arm, 3 with the non-dominant arm), sustained for 12 s each; (**b**) standardized body position in front of a fixed force platform, against which participants produced a maximal voluntary contraction, with the elbow at 90° (i.e., vertical arrow).

**Figure 2 brainsci-12-01537-f002:**
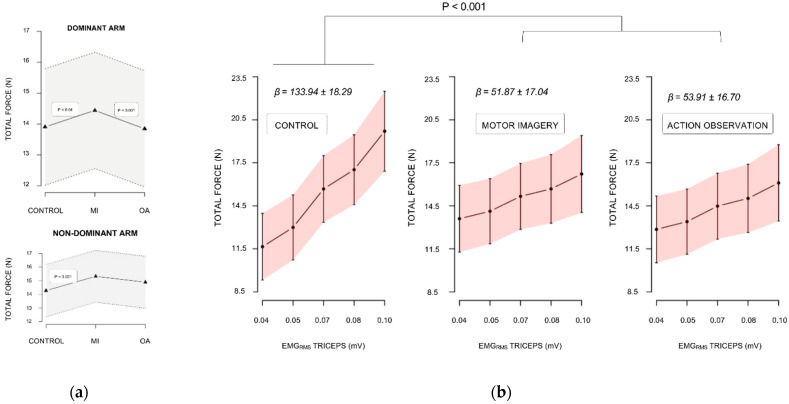
Linear mixed effects analysis of the total force. (**a**) Total force estimates during condition for the dominant (trained) arm and non-dominant (untrained) arms presented with 95% confidence intervals (dotted lines); (**b**) Total force by *triceps brachii* EMG_RMS_ regression slopes during control, motor imagery, and action observation conditions represented by 95% confidence interval (error bars). MI: motor imagery. AO: action observation.

**Figure 3 brainsci-12-01537-f003:**
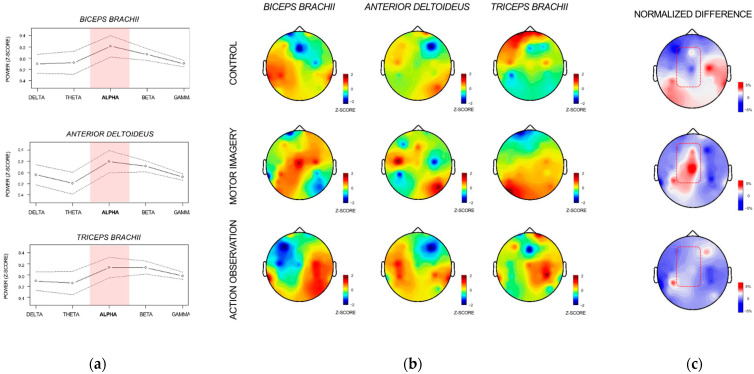
Corticomotor coherence analysis during isometric muscle contraction. (**a**) Standardized coherence power spectrum across frequencies bands for each muscle averaged across experimental conditions presented with 95% confidence intervals (dotted lines); (**b**) imaginary coherence topographies for the alpha (8–12 Hz) frequency range in the EEG sensors-space for all muscles and experimental conditions; (**c**) CMC_RATIO_ topographies used as connectivity indexes of agonist/antagonist corticomuscular connectivity across experimental conditions.

**Figure 4 brainsci-12-01537-f004:**
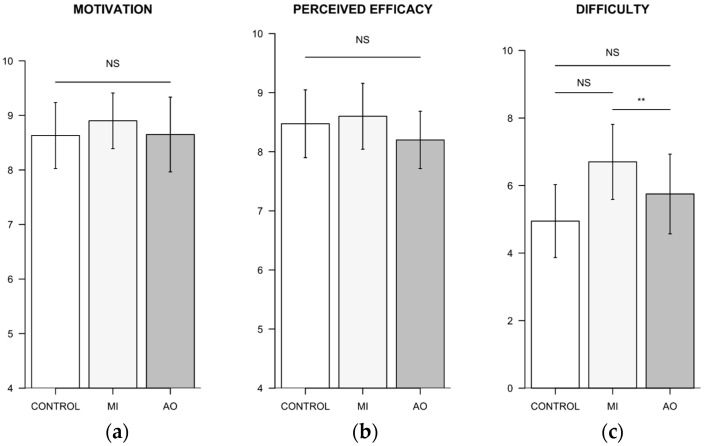
Bar plots obtained from participants’ self-report scores on the 10-points Likert scale. (**a**) Perceived motivation across experimental conditions; (**b**) Perceived efficacy across experimental conditions; (**c**) Perceived difficulty across experimental conditions. Error bars represent 95% confidence intervals. NS: Not statistically significant. MI: motor imagery. AO: action observation. ** *p* < 0.01.

**Table 1 brainsci-12-01537-t001:** Raw total force and EMG_RMS_ data (Mean ± 95% CI) recorded during warm-up trials for each experimental session, for both arms.

**Dominant Arm**			
	**CONTROL**	**MI**	**OA**
Force (N)	16.56 ± 3.97	16.51 ± 2.80	16.88 ± 3.72
EMG_RMS_ *biceps brachii* (mV)	0.10 ± 0.03	0.08 ± 0.02	0.09 ± 0.03
EMG_RMS_ *anterior deltoideus* (mV)	0.72 ± 0.25	0.73 ± 0.24	0.76 ± 0.28
EMG_RMS_ *triceps brachii* (mV)	0.06 ± 0.01	0.11 ± 0.09	0.06 ± 0.01
**Non Dominant Arm**			
	**CONTROL**	**MI**	**OA**
Force (N)	17.18 ± 5.39	15.03 ± 2.79	15.99 ± 3.44
EMG_RMS_ *biceps brachii* (mV)	0.05 ± 0.02	0.05 ± 0.02	0.06 ± 0.02
EMG_RMS_ *anterior deltoideus* (mV)	0.55 ± 0.18	0.62 ± 0.19	0.59 ± 0.17
EMG_RMS_ *triceps brachii* (mV)	0.05 ± 0.02	0.05 ± 0.02	0.10 ± 0.10

## Data Availability

The datasets used and/or analyzed during the current pilot trial are available from the corresponding author on reasonable request.

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
