# Peer review of "Corticomotor Plasticity Underlying Priming Effects of Motor Imagery on Force Performance"

_brainsci, 2022, doi:10.3390/brainsci12111537_

Round 1

Reviewer 1 Report

Even if I consider the paper actually fine, here are some additional suggestions for optional changes:

This interesting paper on the priming effects (in terms of cortical control of reduced agonist/antagonist muscular coactivation) of the motor imagery on force performance has an appropriate title and it provides the readers with novel insights on this topic, possibly exploitable in rehabilitative and assistive technology design and development (the introduction and/or the conclusions could advantageously point at this with short examples for enriching the related paragraphs).  In my opinion, the text can be improved after a language check and an extension of the methodological description, especially focusing on the problem before moving to the experimental design. Additional information that can be retrieved through (a priori and/or post-hoc) power analysis (from the sample size to the effect size) could be the ideal completion of the methodological paragraph from the perspective of readers.

Author Response

Please find response file enclosed.

Reviewer 2 Report

As attached notes

Author Response

Please find response file enclosed.

Round 2

Reviewer 2 Report

In view of the adjustments presented, I consider the manuscript in a condition to be published.